

# Applied phylogeography of *Cyclopia intermedia* (Fabaceae) highlights the need for 'duty of care' when cultivating honeybush

Nicholas C. Galuszynski and Alastair J. Potts

Department of Botany, Nelson Mandela University, Port Elizabeth, Eastern Cape, South Africa

## ABSTRACT

**Background**. The current cultivation and plant breeding of Honeybush tea (produced from members of *Cyclopia* Vent.) do not consider the genetic diversity nor structuring of wild populations. Thus, wild populations may be at risk of genetic contamination if cultivated plants are grown in the same landscape. Here, we investigate the spatial distribution of genetic diversity within *Cyclopia intermedia* E. Mey.—this species is widespread and endemic in the Cape Floristic Region (CFR) and used in the production of Honeybush tea.

**Methods**. We applied High Resolution Melt analysis (HRM), with confirmation Sanger sequencing, to screen two non-coding chloroplast DNA regions (two fragments from the *atpI-aptH* intergenic spacer and one from the *ndhA* intron) in wild *C. intermedia* populations. A total of 156 individuals from 17 populations were analyzed for phylogeographic structuring. Statistical tests included analyses of molecular variance and isolation-by-distance, while relationships among haplotypes were ascertained using a statistical parsimony network.

**Results**. Populations were found to exhibit high levels of genetic structuring, with 62.8% of genetic variation partitioned within mountain ranges. An additional 9% of genetic variation was located amongst populations within mountains, suggesting limited seed exchange among neighboring populations. Despite this phylogeographic structuring, no isolation-by-distance was detected ($p > 0.05$) as nucleotide variation among haplotypes did not increase linearly with geographic distance; this is not surprising given that the configuration of mountain ranges dictates available habitats and, we assume, seed dispersal kernels.

**Conclusions**. Our findings support concerns that the unmonitored redistribution of *Cyclopia* genetic material may pose a threat to the genetic diversity of wild populations, and ultimately the genetic resources within the species. We argue that 'duty of care' principles be used when cultivating Honeybush and that seed should not be translocated outside of the mountain range of origin. Secondarily, given the genetic uniqueness of wild populations, cultivated populations should occur at distance from wild populations that is sufficient to prevent unintended gene flow; however, further research is needed to assess gene flow within mountain ranges.

Corresponding author
Nicholas C. Galuszynski,
nicholas.galuszynski@gmail.com

## INTRODUCTION

The Cape Floristic Region (CFR), located along the southern Cape of Africa, hosts over 9,000 species of flowering plants, of which nearly 70% are endemic to the region (*Goldblatt & Manning, 2002*). With such diversity, it is unsurprising that the Cape flora has a long history of commercial harvesting, cultivation and trade, dating back to seventeenth century European settlers (*Scott & Hewett, 2008*). This includes members of the CFR endemic genus *Cyclopia* Vent., used for the production of a herbal infusion referred to as Honeybush tea (*Hofmeyer & Phillips, 1922*; *Kies, 1951*). The Honeybush industry relies predominantly on raw material sourced from wild populations, but this approach is unable to meet consumer demand and a transition to agricultural production is underway to avoid over-exploitation of natural resources (*Joubert et al., 2011*). This expansion of the cultivated Honeybush sector should reduce harvesting pressure on wild populations and provide opportunities for employment in economically depressed rural communities. However, the underlying distribution and level of genetic diversity present in wild populations is rarely considered during the transition to cultivation plants, and often involves a period of screening individuals for commercially favourable traits (*Hyten et al., 2006*; *Schipmann et al., 2005*). This has been the case for Honeybush, with individuals sourced from multiple populations and species for initial breeding trials (*Joubert et al., 2011*). When establishing cultivated populations, failing to represent the genetic diversity patterns of local populations can potentially place wild populations at risk of contamination by non-local genetic lineages (*Hammer et al., 2009*; *Laikre et al., 2010*). Potential issues of genetic contamination associated with the cultivation Honeybush have recently been raised *Potts (2017a)*. Based on the limited dispersal abilities of *Cyclopia* species (seed dispersed by dehiscent seed pods and ants, and pollen by carpenter bees, *Xylocopa* spp., *Schutte, 1997*), *Potts (2017a)* argues that *Cyclopia* populations are likely to exhibit geographically structured genetic diversity, which may be lost or disrupted if not considered during the transition to cultivation. Here we apply High Resolution Melt analysis (HRM; (*Wittwer et al., 2003*), coupled with confirmation Sanger sequencing (*Sanger, Nicklen & Coulson, 1977*), of chloroplast genetic lineages in the widespread *Cyclopia intermedia* E. Mey to test whether spatial structuring exists in this species.

Genetic diversity provides the basis for evolutionary change, including acclimation and adaptive responses to changes in local environmental conditions (*Pauls et al., 2012*). Genetic diversity is not, however, evenly distributed across a species' range. Rather, demographic history shapes the spatial structuring and extent of molecular variation within a species (*Charlesworth, 2009*; *Excoffier, Foll & Petit, 2009*; *Klopfstein, Currat & Excoffier, 2005*), providing the basis for phylogeography (*Avise et al., 1987*). As biogeographic processes (climate and geomorphic changes) fragment and isolate populations, neutral genetic processes (mutation and drift) lead to genetic divergence among populations (if isolated for sufficient time), producing patterns of phylogeographic structuring. Large interconnected populations are likely to exhibit lower levels of genetic drift (*Kimura & Crow, 1963*) as the large effective population size facilitates the persistence of rare alleles (generated by mutation or migration) at low frequencies (*Tajima, 1989*) and therefore have higher levels

of genetic diversity. Alternatively, small populations are often isolated and prone to genetic drift and inbreeding, reducing overall genetic diversity (*Ellstrand & Elam, 1993*). As argued in *Potts (2017a)*, it is likely that these processes have driven phylogeographic structuring in the CFR given its topographic and environmental heterogeneity—and thus high-levels of genetic differentiation amongst populations is likely the norm, rather than the exception.

These demographic processes, and their impacts on intraspecific genetic diversity, are rarely considered when wild crop or medicinal plants are introduced to an agricultural setting. This has resulted in genetic divergence between neighbouring wild and cultivated populations (*Yuan et al., 2010*; *Otero-Arnaiz, Casas & Hamrick, 2005*), creating opportunities for genetic pollution of wild populations (*Bredeson et al., 2016*; *Bartsch et al., 1999*; *VandenBroeck et al., 2004*; *Millar & Byrne, 2007*) and potentially disrupting species boundaries through the formation of hybrid taxa (*Macqueen & Potts, 2018*; *Lexer et al., 2003a*; *Lexer et al., 2003b*). For example, wild harvesting of the Chinese medicinal plant *Scutellaria baicalensis* (Lamiaceae) has resulted in rapid declines in population sizes and, subsequently, widespread cultivation was promoted to meet demands and reduce harvesting pressure on wild populations (*Yuan et al., 2010*). By screening genetic diversity and structuring of 28 wild and 22 cultivated *S. baicalensis* populations, *Yuan et al. (2010)* demonstrated that although cultivated populations supported similar levels of genetic diversity, they did not represent the phylogeographic structuring of wild populations, as genetic types were widely redistributed outside of their natural ranges. Moreover, if genetically divergent wild and cultivated variants occur within pollination distance from one another, have overlapping flowering times, and are sexually compatible, then gene flow is likely to occur (*Arias & Rieseberg, 1994*; *Bartsch et al., 1999*; *Papa & Gepts, 2003*; *Otero-Arnaiz, Casas & Hamrick, 2005*). Genetic pollution of crop wild relatives has been widely documented since the advent of genetically modified (GM) crop plants, with the majority of globally important GM crops exhibiting hybridization with at least one wild relative (*Ellstrand, Prentice & Hancock, 1999*). It is therefore suggested that 'duty of care' be taken when introducing new crops into agricultural systems, so as to minimise any potential negative ecological effects (*Byrne & Stone, 2011*).

Like the majority of members of the highly-diverse CFR flora, the extent of geographic structuring of chloroplast genetic diversity of members of *Cyclopia* is unknown, and no precautionary limit to the redistribution of seed material exists to guide cultivation. This study therefore sets out to test the postulation put forward by *Potts (2017a)*, that—due to the topographic complexity of the Cape landscape coupled with limited seed dispersal by ants—members of *Cyclopia* will exhibit highly structured chloroplast genetic diversity; such a pattern has already been observed in the preliminary haplotype screening of wild *Cyclopia subternata* Vogel. populations (*Galuszynski & Potts, 2020*). Through a combination of High Resolution Melt analysis (HRM; *Wittwer et al., 2003* and Sanger sequencing *Sanger, Nicklen & Coulson, 1977*) of two non-coding chloroplast DNA (cpDNA) regions (two fragments from the *atpI-aptH* intergenic spacer and one from the *ndhA* intron), an applied phylogeographic approaches is used to describe the levels and distribution of haplotype variation in the widespread and commercially important Honeybush species *C. intermedia*.

The findings of this study provide baseline data for the development of guidelines for the management and protection of wild genetic diversity in *C. intermedia*.

## MATERIALS & METHODS

### Study species

*Cyclopia intermedia* is the most widespread member of the genus (consisting of 23 species; (*Schutte, 1997*), occurring in isolated patches across all major mountain ranges within the eastern CFR (*Schutte, 1997*), sampling locations provided in Fig. 1 and Table 1; further descriptions of these mountain rages are provided in Supplemental Information 1); the species has a broad habitat tolerance occurring at elevations ranging from 500–1,700 m on rocky, loamy, and sandy soils of adequate depth. After fires, it can resprout from a woody rootstock to form a dense shrub (see figures in Supplemental Information 1). This species has poor post-fire recruitment from seed compared to obligate reseeding members of the genus (*Schutte, Vlok & Van Wyk, 1995*). Although *C. intermedia* was not initially targeted for cultivation, it currently accounts for the majority of wild harvested Honeybush (*McGregor, 2017*), and commercial cultivation of this species is on the rise (N. C. Galuszynski. pers. obs., 2018). As such, it is unlikely that genetic material has been widely translocated for this species and opportunities exist to guide future conservation and management of genetic diversity.

### Sample collection and DNA extraction

Over the period of 2015–2018, samples were collected from wild populations across the natural range of *C. intermedia*. Due to many widespread fires across the CFR during this period, many populations were limited to individuals that were either in the process of resprouting or those found in micro-refugia (such as rocky outcrops) where they were protected from fire. Thus, sample size is highly variable among populations (ranging between 1 and 16). In populations with less than 10 plants, all individuals were sampled, in large populations plants were collected across the full extent of the stand with a minimum distance of five meters between sampled individuals. All sampling was approved by the relevant permitting agencies: Cape Nature (Permit number: CN35-28-4367), the Eastern Cape Department of Economic Development, Environmental Affairs and Tourism (Permit numbers: CRO 84/ 16CR, CRO 85/ 16CR), and the Eastern Cape Parks and Tourism Agency (Permit number: RA_0185). Fresh leaf material was collected from the healthy growing tips of individuals and placed directly into a silica desiccating medium for a minimum of two weeks prior to DNA extraction. A modified CTAB DNA extraction approach was adapted from the methods outlined by *Doyle & Doyle (1987)*. Once extracted, genomic DNA was quantified using a NanoDrop 2000c Spectrophotometer (Thermo Fisher Scientific, Wilmington, DE19810r Scientific, USA) and 5 ng/L DNA dilutions were made for PCR amplification and HRM analysis.

### Haplotype detection and sequence alignment

High Resolution Melt analysis involves the gradual heating of PCR products amplified in the presence of a DNA saturating dye. As the double stranded DNA is melted, it dissociates

Galuszynski and Potts (2020), *PeerJ*, DOI 10.7717/peerj.9818

**Table 1  Population location and haplotype assignment.** Population names and mountain range of origin (following the abbreviated format used in fig1,fig4, full names provided in table footer), and GPS coordinates of each population in decimal degree format. The number of accessions sampled per population (N) ranges between 1 and 16 based on the number of individuals found in the field. The number of accessions assigned to each haplotype (A–W) is also provided.

| Population | | Mountain range | Lon | Lat | N | A | B | C | D | E | F | G | H | I | J | K L | M | N | O | P | Q | R | S | T | U V | W |
|---|---|---|---|---|---|---|---|---|---|---|---|---|---|---|---|---|---|---|---|---|---|---|---|---|---|---|
| Anysberg | (AB) | AB | 20.72 | −33.50 | 13 | 13 | – | – | – | – | – | – | – | – | – | – | – | – | – | – | – | – | – | – | – | – |
| Garcia's pass | (GAR) | LB | 21.22 | −33.96 | 15 | – | 15 | – | – | – | – | – | – | – | – | – | – | – | – | – | – | – | – | – | – | – |
| Besemfontein | (BF) | KSB | 21.47 | −33.37 | 10 | – | – | 5 | 5 | – | – | – | – | – | – | – | – | – | – | – | – | – | – | – | – | – |
| Rooiberg | (RB) | RB | 21.57 | −33.68 | 1 | – | – | – | – | 1 | – | – | – | – | – | – | – | – | – | – | – | – | – | – | – | – |
| Swartberg pass | (SMP) | SB | 22.04 | −33.33 | 16 | – | – | – | – | – | 13 | 1 | 1 | 1 | – | – | – | – | – | – | – | – | – | – | – | – |
| Swartberg mountains | (SBM) | SB | 22.38 | −33.38 | 3 | – | – | – | – | – | – | 1 | – | – | 1 | 3 | – | – | – | – | – | – | – | – | – | – |
| Blesberg | (BB) | SB | 22.76 | −33.41 | 3 | – | – | – | – | – | 3 | – | – | – | – | – | – | – | – | – | – | – | – | – | – | – |
| Doring River | (DR) | OUT | 22.31 | −33.88 | 3 | – | – | – | – | – | – | – | – | – | – | 3 | – | – | – | – | – | – | – | – | – | – |
| Prince Alfred's pass | (PAP) | LK | 23.16 | −33.76 | 8 | – | – | – | – | – | 1 | – | – | – | – | – | 2 | – | – | – | – | 2 | 1 | 2 | – | – |
| Haarlem | (HAR) | LK | 23.32 | −33.77 | 9 | – | – | – | – | – | – | – | – | – | – | – | 8 | – | 1 | – | – | – | – | – | – | – |
| Bluehills | (BH) | LK | 23.41 | −33.59 | 5 | – | – | – | – | – | – | – | – | – | – | – | 4 | 1 | – | – | – | – | – | – | – | – |
| Langkloof b | (LKb) | LK | 23.71 | −33.79 | 12 | – | – | – | – | – | – | – | – | – | – | – | 10 | – | – | 2 | – | – | – | – | – | – |
| Langkloof a | (LKa) | LK | 23.79 | −33.78 | 16 | – | – | – | – | – | – | – | – | – | – | – | 15 | – | – | – | 1 | – | – | – | – | – |
| Kouga Wilderness | (KW) | LK | 23.83 | −33.67 | 10 | – | – | – | – | – | – | – | – | – | – | – | 10 | – | – | – | – | – | – | – | – | – |
| Baviaanskloof | (BAV) | B | 24.48 | −33.62 | 11 | – | – | – | – | – | 8 | – | – | – | – | – | – | – | – | – | – | – | – | – | 21 | – |
| Longmore Forest | (LMF) | CC | 25.11 | −33.82 | 5 | – | – | – | – | – | – | – | – | – | – | – | – | – | – | – | – | – | – | – | – | 5 |
| Lady Slipper | (LS) | CC | 25.26 | −33.89 | 16 | – | – | – | – | – | – | – | – | – | – | – | 1 | – | – | – | – | – | – | – | – | 15 |

**Notes.**
Mountain ranges: Anysberg (AB), Klein Swartberg (KSB), Rooiberg (RB), Groot Swartberg (SB), Outeniqua (OUT), Langkloof (LK), Baviaanskloof (B), Cockscomb (CC).
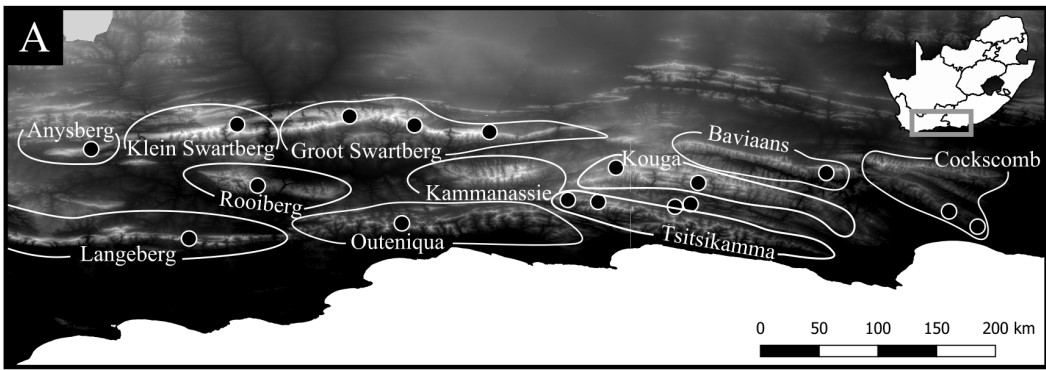

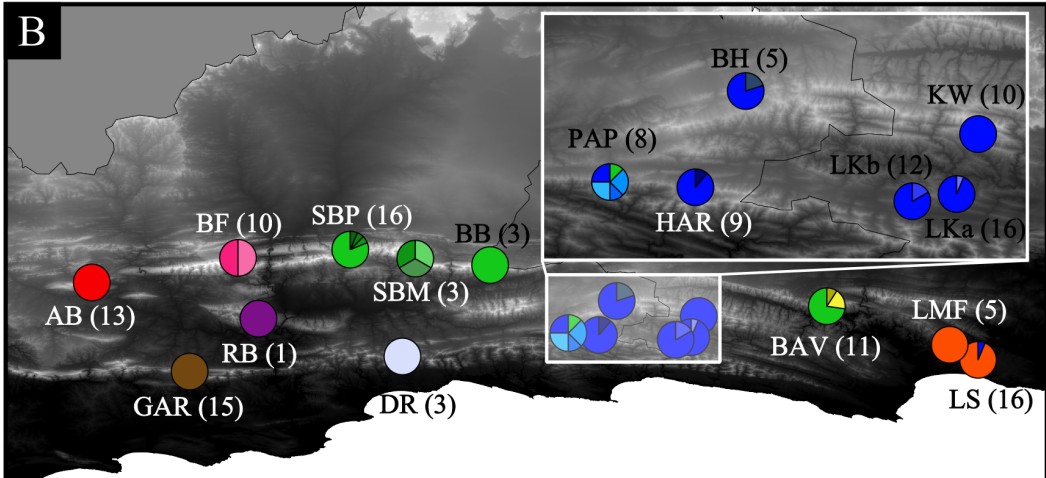

**Figure 1 Geographic distribution of the 17 *Cyclopia intermedia* populations screened for haplotype variation.** (A) High contrast Digital Elevation Map of the mountain ranges of the eastern CFR, including the sampling locations of the *C. intermedia* populations included in the phylogeographic analysis. Note that the Kouga and Tsitsikamma mountain ranges are linked by continuous fynbos habitat and are locally referred to as a single area, the Langkloof. The study area in relation to South Africa is indicated in the inset. (B) Haplotype frequencies (partitioning of circles) and sample sizes (in parenthesis) of the *C. intermedia* populations included in the study, population names follow the description in Table 1. Populations that overlap and obscure the haplotype frequencies are indicated in the inset. The haplotype frequencies follow the same colour scheme as Fig. 2, with haplotypes color coded based on mountain range of origin: red indicates Anysberg (AB), brown is the Langeberg (GAR), pink is the Klein Swartberg (BF), purple is the Rooiberg (RB), green is Groot Swartberg (SBP, SBM, BB), off white is Outeniqua (DR), blue is Langkloof (PAP, HAR, BH, LKa, LKb, KW), yellow is Baviaans Kloof (BAV), and orange is Cockscomb (LMF, LS).

at a rate based on the nucleotide binding chemistry of the double stranded DNA molecule under analysis. As such, different nucleotide sequences usually produce distinct melt curves when plotting the normalised fluorescence differences among samples against temperature (not every unique sequence has a differentiable melt curve). The application of HRM has been demonstrated to be a rapid and cost effective means of detecting haplotype variation in wild *Cyclopia* populations (*Galuszynski & Potts, 2020*) and other plant species (e.g., *Amphicarpaea bracteata* (L.) Fernald, Fabaceae: *Kartzinel et al., 2016*; *Arenaria ciliata* L. and *Arenaria norvegica* Gunnerus, Caryophyllaceae: (*Dang et al., 2012*); *Olea europaea* L.,

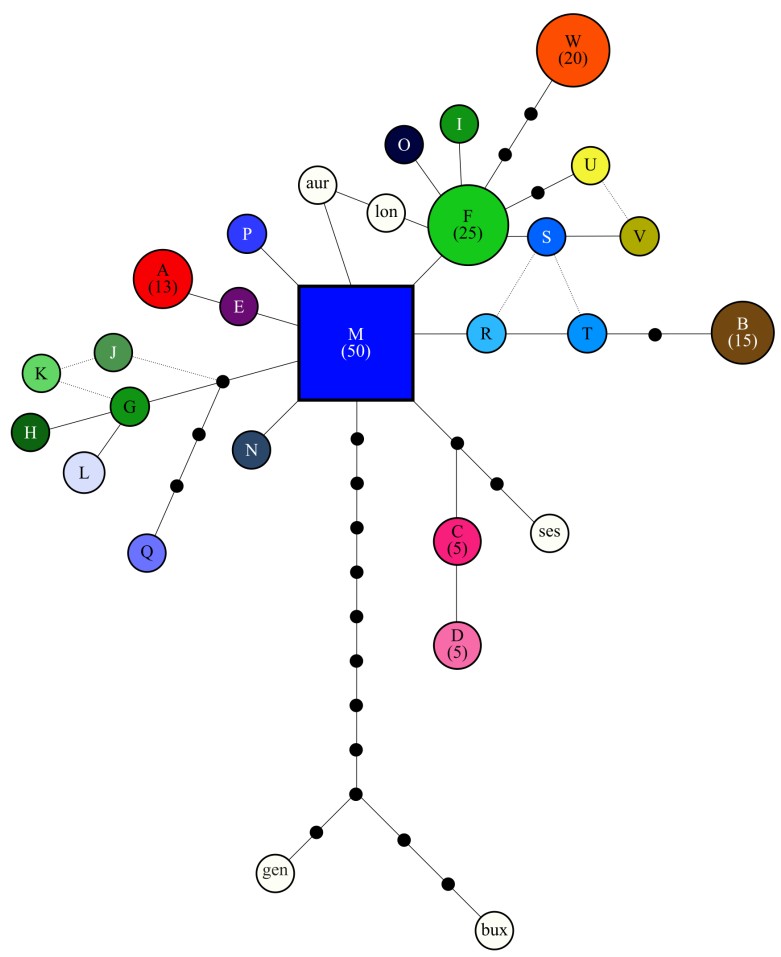

**Figure 2** **Relationship among cpDNA haplotypes, as inferred from the statistical parsimony algorithm.** All haplotypes are colorized based on Fig. 1 and labeled as in Table 1 and the putative ancestral haplotype (M) is indicated as a square. Frequency of haplotypes that were detected five times or more in the study are provided in parenthesis, while the area of circles is scaled based on haplotype frequency. Black circles indicate "missing" haplotypes, while haplotypes connected by a single line differ by one base pair. When the relationship between haplotypes is uncertain a broken line is used. White circles indicate the outgroup taxa: aur = *Cyclopia aurescens*, bux = *C. buxifolia*, gen = *C. genistoides*, lon = *C. longifolia*, ses = *C. sessiliflora*.

Oleaceae: (*Muleo et al., 2009*); and, *Alnus glutinosa* (L.) Gaertn., Betulaceae: *Cubry et al., 2015*).

As the application of HRM to members of *Cyclopia* populations has been described elsewhere (*Galuszynski & Potts, 2020*), only a brief overview of the method is provided here. Two non-coding cpDNA regions (two fragments from the *atpI-aptH* intergenic spacer and one from the *ndhA* intron) were amplified using *Cyclopia* specific primers (provided in Table 2) and subsequently screened for nucleotide variation as detected by HRM curve analysis (i.e., samples with different melt curves). Samples were run in replicates of two, and grouped by population using the well group option in the CFX Manager Software (Bio-Rad Laboratories, Hercules, California, USA) to allow for HRM

**Table 2 Polymerase Chain Reaction primers used to detect haplotype variation across two noncoding cpDNA regions in wild *Cyclopia* intermedia populations.** Primers used for High Resolution Melt analysis are denoted in bold typeface and primers used to amplify the full cpDNA region are denoted in italic typeface. The primers used for unidirectional sequencing of PCR products for HRM cluster confirmation are underlined and italicised. The average number of length of PCR products amplified by the various primer combinations are given in base pairs (bp), followed by the primers nucleotide motif, and the annealing temperature used for PCR.

| cpDNA region | Primer ID | PCR product size (bp) | Sequence (5′- >3′) | Tm (°C) |
|---|---|---|---|---|
| *ndhA intron* | ndhAx1 | 1100 | GCYCAATCWATTAGTTATGAAATACC | 55 |
| | *ndhAx2* | | *GGTTGACGCCAMARATTCCA* | |
| | **MLT_U1** | **350** | **AGGTACTTCTGAATTGATCTCATCC** | **59.0** |
| | **MLT_U2** | | **GCAGTACTCCCCACAATTCCA** | |
| *atpI-atpHintergenic spacer* | atpI | 1100 | TATTTACAAGYGGTATTCAAGCT | 50 |
| | *atpH* | | *CCAAYCCAGCAGCAATAAC* | |
| | **MLT_S1** | **200** | **TGGGGGTTTCAAAGCAAAGG** | **58.8** |
| | **MLT_S2** | | **ATTACAGATGAAACGGAAGGGC** | |
| | **MLT_S3** | **300** | **TTCCCGTTTCATTCATTCACATTCA** | **59.3** |
| | **MLT_S4** | | **CCTTTGCTTTGAAACCCCCA** | |

**Table 3 Polymerase Chain Reaction and High Resolution Melt condition used to screen haplotype variation in wild *Cyclopia* intermedia populations.** Primer annealing temperature (Tm) for the various primer combinations used in this study are provided in Table 2.

| Process | Step | Temperature | Time | Number of cycles |
|---|---|---|---|---|
| PCR Amplification | Initial Denaturing | 95 °C | 2 min | 1 |
| | Denaturing | 95 °C | 10 s | 40 |
| | Annealing + Plate Read | **Primer Tm** | 30 s | |
| | Extension + Plate Read | 72 °C | 30 s | |
| HRM Analysis | Heteroduplex Formation | 95 °C | 30 s | 1 |
| | | 60 °C | 1 min | 1 |
| | HRM + Plate Read | 65–95 °C (in 0.2 °C increments) | 10 sec/step | 1 |

clustering to be conducted on a per population basis following the workflow suggested by *Dang et al. (2012)*. All reactions (PCR amplification and subsequent HRM) took place in a 96 well plate CFX Connect (Bio-Rad Laboratories, Hercules, California, USA) and PCR and HRM conditions are provided in Table 3. All haplotype melt curve grouping was performed on normalized fluorescence differences curves using the automated clustering algorithm of the High Precision Melt software (Bio-Rad Laboratories, Hercules, California, USA) (Tm threshold = 0.05; curve shape sensitivity = 70%; temperature correction = 20).

As HRM analysis does not provide insights into the specific nucleotide motifs under analysis, the haplotype identity of each HRM cluster was confirmed by sequencing a subset of individuals belonging to each HRM cluster per population (PCR amplification following the protocols of *Shaw et al. (2007)*, which are described in Table S1). Since the HRM analysis targeted a subsection of the cpDNA regions investigated, unidirectional sequencing of the full region (using the reverse primers indicated in Table 2), proved sufficient for verifying the sequence motifs amplified by the genus specific HRM primers.

In cases where sample replicates were classified as two different putative haplotypes by HRM clustering, confirmation sequencing of the sample was employed to ensure correct haplotype assignment. Additionally, in cases where HRM clusters represented false negative haplotype assignments (i.e., different haplotypes, as determined from confirmation sequencing, being assigned to the same HRM cluster), all the individuals assigned to the HRM cluster were sequenced for haplotype assignment; this, however, was only required in one instance (described in the results).

Sanger confirmation sequences were assembled using *CondonCode Aligner [v2.0.1]* (Codon Code Corp, http://www.codoncode.com). Each base-call was assigned a quality score using the PHRED base-calling program (*Ewing et al., 1998*). Following this, sequences were automatically aligned using ClustalW (*Thompson, Higgins & Gibson, 1994*) and visually inspected for quality, all small (2–3 bp) indels occurring in homopolymer repeat regions that proved difficult to score were removed. All *C. intermedia* samples that underwent HRM analysis were then assigned the haplotype identity of the HRM cluster they belonged to using a custom *R* script (provided as Supplemental Information 2). The cpDNA regions under investigation are maternally inherited in tandem and not subject to recombination (*Reboud & Zeyl, 1994*), and were therefore combined for subsequent analysis. Haplotype distribution and within populations frequency was mapped using *QGIS [3.2.2]* (*Lacaze, Dudek & Picard, 2018*) (Fig. 1, Table 1).

## Phylogeographic analysis

The genealogical relationship among the *C. intermedia* cpDNA haplotypes was ascertained using Statistical Parsimony (SP) network construction in *TCS [v1.2.1]* (*Clement, Posada & Crandall, 2000*). Five outgroup taxa (*C. aurescens* Kies, *C. buxifolia* (Burm.f.) Kies, *C. genistoides* (L) R.Br., *C. longifolia* Vogel, and *C. sessiliflora* (L) R.Br.) that were previously sequenced to develop the genus specific HRM markers (*Galuszynski & Potts, 2020*) were included in the SP network. Default options were used to build the network, although each indel event was reduced to a single base pair (*TCS* counts each base as an independent evolutionary event). Spatial partitioning of genetic diversity across mountain ranges (mountain range assignment for each population is provided in Table 1) and populations was tested using an Analysis of Molecular Variance (AMOVA; *Excoffier, Smouse & Quattro, 1992*), conducted using the *poppr [v2.8.3]* library (*Kamvar, Tabima & Grünwald, 2014*; *Kamvar, Brooks & Graanwald, 2015*). Isolation By Distance (*Wright, 1943*) was calculated using untransformed geographic distance, as well as the log natural algorithm of geographic distance; the latter accounts for overdispersion in the data resulting from non-linear patterns of genetic divergence among populations (*Rousset, 1997*). The genetic divergence indices used to calculate isolation-by-distance (IBD) included: Jost's D (*Jost, 2008*) (calculated using the *mmod [v1.3.3]* library; (*Winter, 2012*), which measures haplotype differentiation between populations, and Prevosti's distance (*Prevosti, Ocana & Alonso, 1975*) (calculated using the *poppr* library), a measure of pairwise population genetic distance that treats indels as a fifth character state (all indels were reduced to single base events for the same reason described above). Significance of the IBD analyses were tested via

a Mantel tests, with 9999 replicates (*Mantel, 1967*), implemented via the *randtest* function in *ade [v4 1.7]* (*Dray & Dufour, 2007*).

Expected haplotype richness, that corrects for unequal sample sizes among populations, was calculated using the *poppr* function in the *poppr* library, and mean haplotype fixation was calculated for each population using pairwise G"st (*Hedrick, 2005*), calculated in the *mmod [v1.3.3]* library (*Winter, 2012*) and plotted against longitude. This longitudinal trend in haplotype turnover was investigated as the Cape Fold mountains that support *C. intermedia* populations consist of linear arrangement of ranges that are dissected by deep river gorges (Fig. 1 and described further in Supplemental Information 1) that are likely major barriers to seed dispersal. Genetic clustering of populations was estimated from a Neighbour Joining dendrogram constructed from pairwise population genetic distance (*Prevosti, Ocana & Alonso, 1975*) calculated above. Branch support was tested via a bootstrap analysis conducted with 9999 replicates using the *aboot* function in the *poppr* library. All analyses were conducted in *R [v3.5.1]* (*R Core Team, 2018*).

# RESULTS

## High Resolution Melt haplotype detection

High Resolution Melt proved to be an effective tool for detecting haplotype variation in wild *C. intermedia* populations—with a total of 156 individuals screened across 17 populations—detecting 23 haplotypes (confirmed by sequencing a total of 86 and 79 individuals across the two non-coding cpDNA regions respectively). Only one false negative was detected in the study (HRM clustering two different haplotypes, amplified by the MLT S3 - MLT S4 primer combination, as the same putative haplotype), which was resolved by sequencing the *atpI-atpH* intergenic spacer for both individuals assigned to that cluster. (Note that this is why sequencing of a random subset of samples from each HRM curve cluster was conducted—to detect such possible false negatives).

The final merged cpDNA dataset consists of 844 base pairs, 505 from the *atpI-atpH* intergenic spacer and 339 from the *ndhA* intron, with an overall GC content of 28.1%. The data contained 23 polymorphic sites including four indels, nine transversions, and 13 transitions. Nucleotide differences between *C. intermedia* haplotypes are reported in Table 4.

## Phylogeographic analysis

The SP network revealed no distinct haplotype groups. Rather, most of the haplotype variation radiates from an ancestral haplotype (M), located at the center of the network (Fig. 2). Haplotypes exhibited little divergence, many differing by a single substitution. Clustering of populations (based on pairwise genetic distance) in the NJ tree (Fig. 3) was largely consistent with a shared mountain of origin; for example, populations from Langkloof formed a well-supported group with 96.7% bootstrap support. However, notable deviations from this were apparent as genetically unique populations from the Swartberg Mountains (BF and SBM in Fig. 1) formed independent branches outside of the core Swartberg Mountains cluster, and the Baviaanskloof population (BAV in Fig. 1) was grouped in the core Swartberg cluster.

Galuszynski and Potts (2020), *PeerJ*, DOI 10.7717/peerj.9818

**Table 4** **Nucleotide differences among the 23 cpDNA haplotypes detected in wild *Cyclopia* intermedia populations.** The three cpDNA fragments that were individually screened via High Resolution Melt analysis are presented as separate sets of columns, with the nucleotide position within the context of the concatenated haplotype provided for each base pair variation provided. The indels that differentiate haplotypes are reported below the table, with the variation among indel 2a–2e indicated by bold italicised typeface.

| Position | MLT S3 –MLT S4 (*atpI-atpH* intergenic spacer) | | | | | | | | | | | | | | | | | | MLT S1 –MLT S2 (*atpI-atpH* intergenic spacer) | | | | | | MLT U1 –MLT U2 (*ndhA* intron) | | | | | | | | | | |
|---|---|---|---|---|---|---|---|---|---|---|---|---|---|---|---|---|---|---|---|---|---|---|---|---|---|---|---|---|---|---|---|---|---|---|---|
| | 23-29 | 30 | 35 | 38 | 66 | 91 | 99 | 116 | 130 | 136 | 147 | 191 | 192 | 197 | 233 | 272 | 304 | 306 | 321 | 338 | 342–448 | 437 | 442–448 | 462 | 498 | 538 | 569 | 587 | 613 | 677 | 712 | 726 | 739 | 794 | 797–801 |
| Consensus | 1 | T | T | T | A | C | C | G | A | T | T | C | G | A | G | C | A | C | G | G | 2a | G | 3 | T | G | C | G | T | T | G | C | G | G | C | – |
| Haplotype |
| A | . | . | . | . | . | . | . | . | . | . | . | . | . | . | . | . | . | a | . | . | 2b | . | 3 | . | . | . | . | . | . | . | . | . | . | . | – |
| B | . | . | . | . | . | . | . | . | . | . | . | . | . | t | . | . | . | . | . | . | 2d | . | 3 | . | . | . | . | . | . | . | . | . | . | . | 4b |
| C | . | . | . | . | a | . | . | . | . | . | . | a | . | . | . | . | . | . | . | . | 2a | . | 3 | . | . | . | . | . | . | . | . | . | . | . | – |
| D | . | . | . | . | a | . | . | . | . | . | . | a | . | . | . | . | . | . | . | . | 2a | . | 3 | c | . | . | . | . | . | . | . | . | . | . | – |
| E | . | . | . | . | . | . | . | . | . | . | . | . | . | . | . | . | . | a | . | . | 2a | . | 3 | . | . | . | . | . | . | . | . | . | . | . | – |
| F | . | . | . | . | . | . | . | . | . | . | . | . | . | . | . | . | . | . | . | . | 2a | . | 3 | . | . | . | . | . | . | . | . | . | . | . | 4a |
| G | . | . | . | . | . | . | . | . | . | . | . | . | . | . | . | . | . | . | . | . | 2a | . | 3 | . | a | . | . | . | t | . | . | . | . | . | – |
| H | . | . | . | . | . | . | a | . | . | . | . | . | . | . | . | . | . | . | . | . | 2a | . | 3 | . | a | . | . | . | t | . | . | . | . | . | – |
| I | . | . | . | . | . | . | . | . | . | . | . | . | . | . | . | . | . | . | . | . | 2a | . | – | . | . | . | . | . | . | . | . | . | . | . | 4a |
| J | . | . | . | . | . | . | . | . | . | . | . | . | . | . | . | . | . | . | . | . | 2a | . | 3 | . | . | . | . | . | t | . | . | t | . | . | – |
| K | . | . | . | . | . | . | . | . | . | . | . | . | . | . | . | . | . | . | . | . | 2a | . | 3 | . | a | . | . | . | t | . | . | t | . | . | – |
| L | . | . | . | . | . | . | . | . | . | . | . | . | . | . | . | g | . | . | . | . | 2a | . | 3 | . | a | . | . | . | t | . | . | . | . | . | – |
| M | . | . | . | . | . | . | . | . | . | . | . | . | . | . | . | . | . | . | . | . | 2a | . | 3 | . | . | . | . | . | . | . | . | . | . | . | – |
| N | . | . | . | . | . | . | . | . | . | . | . | . | . | . | . | . | . | . | . | . | 2a | . | 3 | . | . | . | . | . | . | . | . | . | a | . | – |
| O | . | . | . | . | . | . | . | . | . | . | . | . | . | . | . | c | . | . | . | . | 2c | . | 3 | . | . | . | . | . | . | . | . | . | . | . | 4a |
| P | . | g | . | . | . | . | . | . | . | . | . | . | . | . | . | . | . | . | . | . | 2a | . | 3 | . | . | . | . | . | . | . | . | . | . | . | – |
| Q | – | . | a | a | . | . | . | . | . | . | . | . | . | t | . | . | . | . | . | . | 2a | . | 3 | . | . | . | . | . | t | . | . | . | . | . | – |
| R | . | . | . | . | . | . | . | . | . | . | . | . | . | . | . | t | . | . | . | . | 2a | . | 3 | . | . | . | . | . | . | . | . | . | . | . | – |
| S | . | . | . | . | . | . | . | . | . | . | . | . | . | . | . | t | . | . | . | . | 2a | . | 3 | . | . | . | . | . | . | . | . | . | . | . | 4a |
| T | . | . | . | . | . | . | . | . | . | . | . | . | . | . | . | t | . | . | . | . | 2a | . | 3 | . | . | . | . | . | . | . | . | . | . | . | 4b |
| U | . | . | . | . | . | . | . | . | . | . | . | . | . | . | . | t | . | . | . | . | 2e | . | 3 | . | . | . | . | . | . | . | . | . | . | . | 4a |
| V | . | . | . | . | . | . | . | . | . | . | . | . | . | . | . | . | . | . | . | . | 2e | . | 3 | . | . | . | . | . | . | . | . | . | . | . | 4a |
| W | . | . | . | . | . | . | . | . | . | . | . | . | . | . | . | . | . | . | . | . | – | . | 3 | . | . | . | . | . | . | . | a | . | a | . | 4a |
| *C. aurescens* | . | . | . | . | . | . | . | . | . | . | . | . | . | . | . | g | . | . | . | . | 2a | . | 3 | . | . | . | . | . | . | . | . | . | . | . | |
| *C. buxifolia* | . | . | . | c | . | . | t | t | c | g | . | t | . | . | . | . | . | . | a | . | 2a | a | 3 | . | . | . | . | c | g | t | . | . | . | . | – |
| *C. genistoides* | . | . | . | c | . | . | . | t | c | g | . | t | . | . | . | . | . | . | a | . | 2a | a | 3 | . | . | . | t | a | c | . | . | . | . | . | – |
| *C. longifolia* | . | . | . | . | . | . | . | . | . | . | . | . | . | . | . | g | . | . | . | . | 2a | . | 3 | . | . | . | . | . | . | . | . | . | . | . | 4a |
| *C. sessiliflora* | . | . | . | . | a | . | . | . | . | . | . | . | . | . | . | g | . | . | . | . | 2a | . | 3 | . | . | . | . | . | . | . | . | . | . | . | 4b |

**Notes.**

1 = TAAAAAT, 3 = TATCTAA, 4a = TT—, 4b = TTTTC, 2a = TACAGATGAAACGGAAGGGGCTTCGTTTTTTGAATCCCTATCTAAATTTACAGTAACAGGGCAAA, 2b = TACAGAT-GAAAGGGAAGGGGCTTCGTTTTTTGAATCCCTATCTAAATTTAAAGTAACAGGGCAAA, 2c = TACAGATGAAACGGAAGGGGCTTCGTTTTTTGAATCCCTATCTAAATTTATAGTAACAGGGCAAA, 2d = TACAGATGAAACGGAAGGGGCTTCGTTTTTTGAAAACCTATCTAAATTTACAGTAACAGGGCAAA, 2e = TACAGATTAAACGGAAGGGGCTTCGTTTTTTGAATCCCTATCTAAATT-TACAGTAACAGGGCAAA,
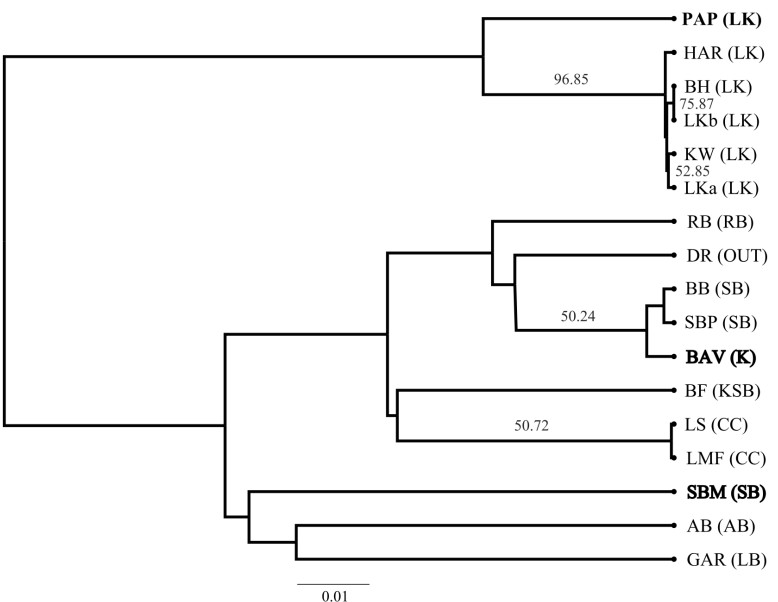

**Figure 3** **Neighbour Joining tree of *Cyclopia intermedia* grouping based on pairwise population genetic distance.** Branches with greater than 50% bootstrap support are labeled and populations that deviate from being grouped with population occurring from the same mountain ranges are denoted in bold typeface; Baviaanskloof (BAV), and Swartberg mountains (SBM), originating from the Groot Swartberg mountains. Population and mountain range naming follows that of Table 1.

**Table 5** **AMOVA results indicating significant structuring of wild *Cyclopia intermedia* populations across mountain ranges.** The degrees of freedom, sum of squares of deviation (SSD), and percentage of genetic variation (% variation) accounted for are provided for the three levels of sample grouping, and the global Fst from the AMOVA is provided.

|  | Degrees freedom | SSD | % variation | Fst |
|---|---|---|---|---|
| Between mountains | 9 | 298.6 | 62.79** | 0.704*** |
| Between populations within mountains | 7 | 21.54 | 8.99 |  |
| Within samples | 139 | 123.32 | 28.22 |  |

**Notes.**
** $p < 0.01$,
*** $p < 0.005$.

Genetic divergence across mountain ranges was further supported by the AMOVA analysis (results reported in Table 5), detecting a significant trend ($p < 0.05$) for genetic diversity to be structured among mountain ranges, accounting for 62.8% of genetic variation, with an additional 9.0% structured among populations within mountain ranges. No IBD was detected for any of the population differentiation measures ($p > 0.05$ across all measures of geographic distance). Plots of expected haplotype richness and G''''st against longitude suggest that edge populations of *C. intermedia* tend to be fixed for a small number locally unique haplotypes (Fig. 4).

We note that outgroup samples nest within the statistical parsimony network. This may be due to a recently shared common ancestor, incomplete lineage sorting or chloroplast

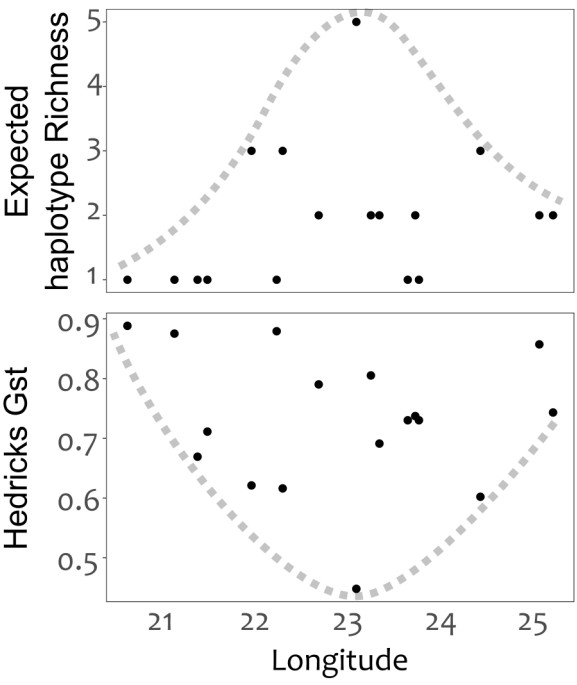

**Figure 4  Haplotype diversity (expected haplotype richness) and differentiation (Hedrick's Gst) measures plotted against the longitudinal position (decimal degrees) of populations.** Broken lines were manually drawn around the distribution of points.

capture (i.e., hybridization). Identifying which driver or drivers gave rise to this observed pattern requires further sampling of species, populations and genomes—sampling the nuclear genome is crucial for identifying hybridization events. At present, little is known about the proclivity for hybridization amongst the species in this genus.

## DISCUSSION

Although *Cyclopia intermedia* is the most widely distributed member of its genus (occurring across all major mountain ranges within the eastern CFR), populations of this species are generally found in localized low density patches on south facing slopes. As stated above, wild harvesting of *C. intermedia* still forms the bulk of the Honeybush industry in the Eastern Cape of South Africa (*McGregor, 2017*), but the recent transition to agricultural production of Honeybush has seen increased cultivation of *Cyclopia* species (*Joubert et al., 2011*), including *C. intermedia*. With no information regarding the spatial structuring of wild *Cyclopia* genetic diversity available, no general rules exist to guide the translocation of seed and seedlings. This poses a potential threat of exposing wild populations to non-local genetic lineages through pollen and seed flow. The escape of non-local genetic lineages may place wild *Cyclopia* at the risk of genetic pollution (*Laikre et al., 2010*; *Potts, 2017a*). The phylogeographic patterns detected for *C. intermedia* (discussed below) support the concerns raised by *Potts (2017a)*, and should provide preliminary guidance to the management and protection of *Cyclopia* wild genetic resources.

## Phylogeography of *C. intermedia*

The phylogeography of *C. intermedia* was explored using HRM coupled with sequencing to screen haplotype variation across two non-coding cpDNA regions. High levels of haplotype turnover among populations and genetic structuring was detected. Phylogeographic work in the CFR has, for the most part, detected phylogeographic structuring of biota regardless of taxonomy or life form (reviewed in *Lexer et al., 2013*; *Tolley et al., 2014*). Unfortunately, few widespread plant taxa have been targeted for phylogeographic work in the region, and even less attention has been given to eastern CFR species. Two studies of widespread CFR plant taxa that naturally occur in the western and eastern CFR (*Tetraria triangularis* (Boeck.) C.B.Clarke: *Britton, Hedderson & Verboom, 2014*); and *Protea repens* (L.) L.: *Prunier et al., 2017*) do, however, highlight the potential for population divergence to occur at a range of spatial scales, including over relatively short distances in heterogeneous landscape of the CFR.

Topographic complexity has been widely cited as an important driver of population isolation in the CFR and adjacent biomes (*Britton, Hedderson & Verboom, 2014*; *Potts et al., 2013a*; *Potts, Hedderson & Cowling, 2013b*; *Prunier et al., 2017*; *Potts, 2017b*; *Prunier & Holsinger, 2010*). Disjunct population distributions in the CFR has been attributed to Pleistocene climate cycles fragmenting populations into climate refugia (*Britton, Hedderson & Verboom, 2014*; *Potts, 2017b*; *Potts et al., 2013a*) as vegetation dynamics in the region shifted in response to changes in rainfall seasonality (*Bar-Matthews et al., 2010*; *Chase & Meadows, 2007*). Furthermore, since seed dispersal is generally limited to short distances in CFR taxa long distance dispersal is unlikely to be responsible for the distribution of populations across adjacent mountain ranges or drainage basins.

In the case of *C. intermedia*, topographic complexity appears to have played a pivotal role in shaping the distribution of genetic diversity, with 62.8% of haplotype variation structured based on mountain ranges (AMOVA), and multiple cases of complete haplotype turnover occurring across adjacent ranges (Fig. 1, Table 1). This was most dramatic across the Anysberg, Besemfontein and Swartberg Pass populations that, despite their close proximity, shared no haplotypes. These patterns of intraspecific divergence are similar to those of *T. triangularis* (*Britton, Hedderson & Verboom, 2014*) and *P. repens* (*Prunier et al., 2017*), where populations sampled from these mountains exhibited high levels of divergence. The deeply incised gorges and expanses of inhospitable lowland habitat separating mountain ranges (see Supplemental Information 1) are barriers dispersal in the CFR, promoting vicariance among upland populations that may have once been more connected during past climate conditions. Additionally, the vegetation composition of the eastern CFR may have seesawed between C3 Mediterranean shrub-lands (that support *C. intermedia*) and alternative woodlands and/or C4 grasslands (*Bar-Matthews et al., 2010*; *Cowling & Lombard, 2002*). While this would have fragmented populations across mountain ranges, as described above, it may have also created opportunities for admixture among populations within mountain ranges and perhaps even across mountain ranges via habitat corridors, especially in the eastern reaches of species ranges. This may explain the widespread distributions of haplotypes M and F in the eastern *C. intermedia* populations (Fig. 1). These haplotypes, while dominant in the Langkloof and Groot Swartberg mountains respectively

(suggesting some degree of connectivity among populations within these ranges), are also found in the eastern most populations located in the Cockscomb (haplotype M) and Baviaanskloof (haplotype F) mountains that both support unique local haplotypes that make them genetically distinct. Considering the slow mutation rate of the chloroplast genome (*Schaal et al., 1998*), the presence of unique haplotypes at low frequencies within populations should be viewed as an additional level of genetic structuring in *C. intermedia*, with 9% of haplotype variation localized in populations within mountain ranges.

Of the 23 haplotypes detected in *C. intermedia*, four are shared among populations (Table 1), possibly suggesting limited dispersal among and within mountain ranges. These rare haplotypes may result from mutations generated in-situ, with limited migration among adjacent populations preventing rare haplotypes from becoming more widespread. Short distance seed dispersal by ants, where 25 m is considered a long distance dispersal event (*Gómez & Espadaler, 2013*), coupled with poor recruitment, would be sufficient to maintain isolation among populations, promoting genetic divergence over short distances. Limited seed dispersal capability has frequently been associated with genetic differentiation among populations of CFR taxa in the past (*Britton, Hedderson & Verboom, 2014*; *Lexer et al., 2003a*; *Lexer et al., 2003b*; *Potts et al., 2013a*; *Zietsman, Dreyer & Jansen Van Vuuren, 2009*). *Britton, Hedderson & Verboom (2014)* detected a number of instances of near complete haplotype turnover among neighboring populations of the high elevation sedge *T. triangularis* sampled from the eastern CFR. While, in the case of the Little Karoo endemic *Berkheya cuneata* (Asteraceae), populations occurring in the western and eastern sub-basins of the Gouritz basin (located between the Groot Swartberg and Outeniqua mountains) have been isolated for a sufficient period to produce highly divergent west and eastern cpDNA lineages (*Potts et al., 2013a*). Limited seed dispersal across environmental and physical barriers (expanses of unsuitable lowland habitat in the case of *T. triangularis* and the Rooiberg mountain in the case of *B. cuneata*) was evoked as possible mechanisms for prolonged population isolation. We postulate that the distribution patterns of rare haplotypes in *C. intermedia* may be a consequence of the same process and represents an important aspect of *Cyclopia* wild genetic diversity in need of protection.

## Distribution patterns of haplotype richness

The lack of IBD (*Rousset, 1997*) and star-burst SP network (*Charlesworth, 2003*) suggests that C. intermedia has experienced a non-linear processes of population isolation and subsequent genetic divergence (i.e., not a single direction stepping stone colonization path), with haplotype M identified as a putative ancestral form due to its central position in the network (*Cann, Stoneking & Wilson, 1987*; *Templeton, Routman & Phillips, 1995*). Some consequences of this are that co-occurring haplotypes have as many, if not more, nucleotide differences separating them than haplotypes originating from geographically distant populations (Fig. 4). This results from novel haplotypes being generated directly from the ancestral form, rather than accumulating mutations along a colonization path (Fig. 2), thus no spatially structured haplotype groups were detected in the SP network and haplotypes located in edge populations are not more genetically differentiated than populations from the center of the species range. For instance, haplotypes A, E and C,
while occurring in different mountain ranges, are genetically more similar to haplotype M than haplotypes L, O, Q, S, and T that occur within the same mountain range as haplotype M (Figs. 1 and 2). The non-linear pattern of population isolation (described above) has a marked impact on within population genetic diversity (*Excoffier, Foll & Petit, 2009*). Populations located in the geographic source of a species' range, (i.e., where the ancestral haplotypes are generally dominant, such as the Groot Swartberg and Langkloof in the case of *C. intermedia*) are likely to have reduced impacts of drift, due to a larger effective population size and long term persistence, promoting the accumulation of genetic diversity (*Excoffier, Foll & Petit, 2009*). This is evident in C. intermedia, with the longitudinal center of the species range having higher expected haplotype richness (Fig. 4). In contrast, populations at the edge of a species range may exhibit reduced genetic diversity (*Klopfstein, Currat & Excoffier, 2005*) and increase fixation of unique local haplotypes (Fig. 4). Novel mutations and rare haplotypes become rapidly fixed in edge populations that are prone to genetic bottlenecks resulting from founder effects or population fragmentation and isolation due to climate changes (*Klopfstein, Currat & Excoffier, 2005*). The potential role of Pleistocene climate instability in isolating C. intermedia populations has been discussed, but we reiterate here the potential role of vicariance in isolating of these genetically distinct edge populations, such as the Anysberg, Garcias Pass, Besemfontein and Lady Slipper populations.

## Management of *C. intermedia* genetic diversity

The goal of identifying historically isolated sets of genetically similar populations is that these populations can then be viewed as distinct management or conservation units—termed "evolutionarily significant units" (*Moritz, 1994*). For example, evolutionarily significant units are restricted to clearly defined drainage basins in some plant species (*Potts et al., 2013a*; *Potts, Hedderson & Cowling, 2013b*). Such units may be especially valuable for guiding the translocation of genetic material in *C. intermedia,* and *Cyclopia* as a whole.

Unfortunately, the biogeographic history of *C. intermedia* described here has not partitioned genetic variation into neat geographic units. And despite the coarse pattern of genetic diversity being structured within mountain ranges—populations tend to support unique haplotypes (Fig. 1, Table 1) that make them genetically distinct (e.g., the Prince Alfred's Pass and Swartberg Mountains populations, Fig. 3) and in need of individual management and protection. Furthermore, a single study sampling only two non-coding regions cannot be considered to represent the entirety of a species genetic diversity. Thus, the fact that 13 of the 17 populations screened here supported unique haplotypes suggests that grouping neighbouring populations as genetically homogenous spatial units for conservation and management would poorly represent the species' cpDNA genetic diversity. Rather than relying on a few *Cyclopia* populations to represent the genetic diversity, or evolutionary significance, of broad areas, all wild populations should be safeguarded from potential genetic pollution to some degree.

### Recommendations for 'duty of care' of *C. intermedia* genetic diversity

Currently, the Honeybush industry relies on raw material sourced predominantly from wild populations, with an estimated 85% of wild harvested Honeybush collected from *C. intermedia* populations (*McGregor, 2017*). With a rise in popularity of natural products in recent years, the Honeybush industry has seen a surge in consumer demand (*Joubert et al., 2008*; *Joubert et al., 2011*), placing these wild populations under additional harvesting pressure. Agricultural cultivation has therefore been encouraged as a means of protecting *Cyclopia* from unsustainable harvesting activities.

The initial Honeybush boom in the 1990s (*Joubert et al., 2011*; *Du Toit, Joubert & Britz, 1998*) spurred on investigations into the cultivation potential of various species of *Cyclopia*, including *C. intermedia*. These investigations involved outcrossing experiments, resulting in successful interspecific crossing (*de Lange & von Mollendor, 2006*) and the introduction of cultivated material into wild populations (N. C. Galuszynski. pers. obs., 2016; S Nortje, pers. com., 2019). Furthermore, Honeybush cultivation has been encouraged to take place near to wild *Cyclopia* populations (*Jacobs, 2008*) that, based on the evidence presented here and elsewhere (*Galuszynski & Potts, 2020*), are likely to be genetically distinct—as previously suspected (*Potts, 2017a*; *Schutte, 1997*). Commercial populations are therefore likely to pose a threat to the genetic integrity of wild *Cyclopia* if gene flow occurs.

Cultivation, as well as interest in augmentation of wild *C. intermedia* populations, is on the rise. The high levels of interpopulation haplotype turnover reported here, most of which coincides with transitions between mountains ranges, should be used to guide future conservation and management of *C. intermedia* seed material—specifically, seed material should be kept within mountain ranges and cultivation should never be used to replace wild populations. While further work is required to describe gene flow patterns within mountain ranges, particularly pollen flow. What can be inferred from the results presented here and existing literature, is that gene flow is likely to be limited to relatively short distances. Currently it is assumed that pollen flow is likely to be limited to a 6 km radius (representing the maximum foraging distance of *Xylocopa* species, (*Pasquet et al., 2008*), the common pollinators of Honeybush, (*Schutte, 1997*) and seed dispersal being limited to under 180 m (globally the longest measured seed dispersal event by ants, *Gómez & Espadaler, 2013*. Honeybush cultivation should therefore operate at a localized scale and facilitate in preserving the genetic uniqueness of *Cylopia* populations within mountain ranges.

## ACKNOWLEDGEMENTS

We would like to acknowledge all those that have made this research possible including: those that facilitated with sampling activities (Gillian McGregor and her students, Shayla Tricam and Jonathan Silverman), Anneslise Schutte-Vlok for her assistance with species identification, all landowners and managers that allowed sampling on their property, and L L Dreyer, F Ojeda, F Forest, and two anonymous reviewers for their comments that greatly improved the clarity of this manuscript.

### Funding

This work was supported by the National Research Fund of South Africa (Grant No. 99034, 95992, 114687) and the Table Mountain Fund (Grant no. TM2499). The funders had no role in study design, data collection and analysis, decision to publish, or preparation of the manuscript.

### Grant Disclosures

The following grant information was disclosed by the authors:
National Research Fund of South Africa: 99034, 95992, 114687.
The Table Mountain Fund: TM2499.

### Competing Interests

Alastair J. Potts is an Academic Editor for PeerJ.

### Author Contributions

- Nicholas C. Galuszynski conceived and designed the experiments, performed the experiments, analyzed the data, prepared figures and/or tables, authored or reviewed drafts of the paper, and approved the final draft.
- Alastair J. Potts conceived and designed the experiments, analyzed the data, authored or reviewed drafts of the paper, and approved the final draft.

### Field Study Permissions

The following information was supplied relating to field study approvals (i.e., approving body and any reference numbers):

All sampling was approved by the relevant permitting agencies, Cape Nature (Permit number: CN35-28-4367), the Eastern Cape Department of Economic Development, Environmental Affairs and Tourism (Permit numbers: CRO 84/ 16CR, CRO 85/ 16CR), and the Eastern Cape Parks and Tourism Agency (Permit number: RA_0185).

### DNA Deposition

The following information was supplied regarding the deposition of DNA sequences:

The sequences for the non-coding chloroplast regions are available at GenBank: MN930803–MN930855 and MN930856–MN930919.

### Data Availability

The sample to haplotype assignment of accessions included in the phylogeographic analysis is available at Figshare: Galuszynski, Nicholas (2020): Cyclopia phylogeography: HRM cluster to haplotype assignment. figshare. Dataset. https://doi.org/10.6084/m9.figshare.11370465.v1.

### Supplemental Information

Supplemental information for this article can be found online at http://dx.doi.org/10.7717/peerj.9818#supplemental-information.

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
