# Peer review of "Applied phylogeography of Cyclopia intermedia (Fabaceae) highlights the need for ‘duty of care’ when cultivating honeybush"

_PeerJ, doi:10.7717/peerj.9818_

## Round 0.1 · original submission · Major Revisions

Please consider suggestions by the two reviewers included below. Crucial information that should be included is to detail methods, for example in results the figure shows the outgroups and they were not mentioned previously. AMOVA methods are explained, however they are not presented in results. In addition, results should be considered to propose priority based in the phylogeographic network.

Reviewer 1 ·

Basic reporting

The authors present a well-written manuscript with a well-referenced introduction that suitably sets the scene for the study. The structure of the paper confirms to PeerJ standards.

Overall the figures are good and present the data well. However, there are a few errors in the captions (detailed below) and I found the colours used for haplotypes P and M are very similar and difficult to distinguish.

The raw data produced is supplied in Tables 2 and 4 and has also been deposited in GenBank. Additional information on the PCR protocol and the custom R script that were used is provided in the Supplemental Files.

Minor comments
Ln 74 – Change ‘I’ to ‘we’ (since there are two authors).

Ln 222- Change ‘all indels were reduced to single base pairs’ to ‘each indel event was reduced to a single base pair’

Ln 258 - Change ‘atpI-aptH intergenic spacer’ to ‘atpI-atpH intergenic spacer’

Ln 283 – should G’’’’’ts be Gst?

Ln 298 – Change (Potts, 2017a) to Potts (2017a)

Ln 351 - Change ‘I’ to ‘we’ (since there are two authors).

Ln 376 – Change to ‘Novel mutations and rare haplotypes become rapidly fixed in populations….’

Ln 433 – Acknowledgments – Change ‘me’ to ‘us’ (since there are two authors).

Ln 478 – The title of the Dray reference should start with ‘The’

Ln 498 – This reference is missing the journal.

Ln 518 – insert space between ‘GST’ and ‘and’

Ln 559 - Restio capensis should be in italics.

Ln 665 - Ambystoma tigrinum should be in italics and be followed by only one full stop.

Figure 1 caption – should ascensions be accessions?

Figure 2 caption – Change – ‘haplotypes connected by a single line differ by a one base pair’ to ‘haplotypes connected by a single line differ by one base pair’

Figure 4 caption – change demoted to denoted.

Table 3 caption – italicize C. intermedia

Table 4 – delete the word ‘various’

Experimental design

The research presented fits within the scope of PeerJ. The study has clear aims and examines these with appropriate sampling and analyses. The methods are clearly explained, although I would like to see details of the outgroup samples added to the ‘Sample collection and DNA extraction’ section. The use of sequencing to confirm the HRM clusters and determine the level of false negative haplotype assignments is particularly commendable and it gives confidence in the quality of the data.

Validity of the findings

Overall, the conclusions are appropriate given the results and link back to the original research question in the introduction.

I have one question about the interpretation of the network. I found it interesting that three of the outgroup species (C. aurescens, C. longifolia and C. sessiflora) have haplotypes that, in the network, are nested within the genetic diversity detected in the study species. Are these species known to hybridise with the study species? Could chloroplast capture from these, or other closely-related species, be the source of some of the rare local haplotypes found in your target species? Chloroplast capture has been found in many overseas studies of plant phylogeography. I don’t think it would change the interpretation of the results or the conclusions but it is something that should be considered.

Additional comments

Overall a good solid study and a nicely written paper. I particularly like that clear conservation recommendations are provided.

Reviewer 2 ·

Basic reporting

I think it is an important contribution because adds useful information about phylogeographic patterns of Cyclopia intermedia, used to produce the Honeybush tea. This species is endemic from the Cape Floristic Region (CFR), located along the southern Cape of Africa. Since the cultivation of this species is relatively recent, opportunities exist to guide future conservation and management based on the retrieved genetic patterns.
However, I consider there are some aspects of the manuscript that should be revised.
Authors should revise considerably the English editing. I think some ideas are confused because of wording. In some places you used "I" instead of "We", please check.
In general terms I consider the introduction is sufficient and clear, the manuscript are appropriately referenced and is also well structured.
Authors do not mentioned if they have deposited or will deposit the sequences in the Genebank; they should do it.

Experimental design

I think the research question is well defined and the knowledge gap is well formulated. The hypothesis should be improved and Materials and Methods should include more detailed information about some issues.

Authors hypothesize that due to limited seed dispersal by ants, Cyclopia species will exhibit highly structured chloroplast genetic diversity as was observed in a previous study of another wild Cyclopia species. In other words, the hypothesis is raised in relation to an autoecological characteristic of the species. On the other hand, the topographic complexity and climatic heterogeneity of the study area are also mentioned in the introduction, but authors did not present an integrated hypothesis considering both factors (i.e. autecology and landscape and environmental characteristics). Thus, I suggest developing a hypothesis that includes both elements. Furthermore, I suggest including in Materials and Methods a description of the study area, since it is not mentioned how high these mountains are, which is the direction of the mountain ranges, and how is the distribution of Cyclopia intermedia within the mountain range. It is not clear for me if there could be altitudinal corridors for the species? Please, explicit how the topography could be a possible barrier for Cyclopia intermedia?
I consider that this will contribute to a better discussion of the results. For example, in the Discussion section the different basins and sub-basins are mentioned and it is difficult to follow it. I suggest including a DEM map (modifying Fig. 1) indicating the location of each basin, mountains, etc. to better understand and discuss the retrieved genetic patterns.

Materials and Methods
Please introduce better the focal species. Is it an herb? a shrub?
Please specify the distance among sampled individuals and the minimum and maximum distance among sampled localities.
Author say that they amplified three non-coding chloroplast DNA loci. However, as you say in line 221, chloroplast DNA are maternally inherited and in tandem, thus the chloroplastidial DNA molecule is considered a single locus. In addition, in my opinion you amplified two non-coding regions (the atpI-aptH intergenic spacer and the ndhA intron), not three as is mentioned in the manuscript; the atpI-aptH intergenic spacer was amplified in two parts. Please modify.
Considering phylogeographic analyses, I think with this dataset authors could go further in unravelling the evolutionary history of the species, for example including demographic analyses. But, for the question posed, inferring haplotypes genealogy, their spatial distribution, and characterize the spatial genetic structure may be enough. The comment I have in this regard is in relation to the use of AMOVA. In this analysis, you define a priori groups of populations; so while partitioning can be significant, perhaps there is another barrier that further explains the genetic pattern obtained. Thus, I suggest to include some analysis such as BASP or SAMOVA where the population groups are identified a posteriori. Given that the patterns found were not expected, this type of analysis can help to infer some factor / barrier not previously considered.
Line 187-188 – How many replicates per sample?
Line 205-207- Please, inform how common was this situation
Line 226 - Did sequences retrieved many indels? Why indels were not coded? As suggested for example by Simmon and Ochoterena 2000, Gaps as Characters in Sequence-Based Phylogenetic Analyses
Line 237 - But you don’t have alleles. You mean, haplotypes? It is necessary to give more detail about the genetic distances and the richness estimator used.
Why did not use a genetic distance that consider nucleotide differentiation and haplotype frequency among populations? They are implemented in Arlequin or DNAsp and sure in R.
Line 239 - It takes indels as a fifth state?
Line 241 - Which analyses? Do you mean that significant association between these genetic and geographic distances were tested via Mantel Test? Please reword this sentence.
Line 244 - Did you correct this estimation based on sample sizes? This is important because of the difference in the sampling sizes among populations.
Line 247 - Why did you consider only the longitude? And the latitude? Probably you can better explain how the study area and the area of distribution is to a better understanding

Validity of the findings

RESULTS
I consider results are not well stated and that there are some inconsistencies with the methodology.
In the Result section, Phylogeographic analysis, please include a description of haplotypes geographical distribution. I recommended to characterize the spatial distribution of haplotypes, while mentioning how genealogically close they are. This makes easier to build the underlying biogeographic scenario.
Haplotype network include out-groups as shown in Fig. 2, but it was not mentioned either in M&M or discussion. I suggest taking them out.
Could you please show the AMOVA results (FCT, FST, FIS, p values? In a table in the main manuscript or as supplementary material.
Line 258 – Please put “regions” instead of “loci”
Line 259 - ndh or ndhA, please uniform throughout the manuscript
Line 259-260 Please move the line about haplotype spatial distribution to M&M
Line 287 – Among instead of within
Line 288 - I suggest 9.0% structured among populations within mountain ranges.
Fig. 1. The study area it is not clearly indicated for me. What is it the white map? An amplification of the white are in the Africa map? What is obscure haplotypes? Please explain better. I also recommend to inset a picture showing the mountain ranges more clearly.
Fig. 2. Haplotype network include out-groups as shown in Fig. 2, but it was not mentioned either in M&M or discussion. I do not see any broken line as is mentioned in figure legend

DISCUSSION AND CONCLUSION
Authors conclude that all wild populations should be safeguarded from potential genetic pollution to some degree. Of course that this would be the best option, but I think author should propose an order of priority based on their study. For example considering haplotype diversity, private haplotypes, etc. Moreover, based on Fig. 1, it is evident that to the western side of the distribution range, there would be evidence of isolation among populations, but toward the eastern side there is some evidence of a greater connectivity among some populations. I think this pattern has management implications. I consider the conclusion could be more connected with the results obtained.

In the discussion section, basins and mountain range, are mentioned but they have not been well presented in the previous sections, so it is a little difficult to follow the reasoning of the discussion. As I mentioned above it is necessary to introduce better the geography and the biogeography of the study area. Including a figure where different mountain ranges, basins, etc be shown.

Line 294-297 I suggest moving these lines to M&M
Line 338-339 - Could you give a putative explanation for this pattern?
Line 370 – What process? Please, explain better what would be a physical barrier for C. intermedia…This idea should be also presented in the introduction
Line 379-380 -This could be a consequence of secondary contact area..
Line 405 - Probably this idea should be also proposed in the introduction, in the hypothesis

---

## Round 0.2 · accepted · Accept

I appreciate your effort in considering all issues and suggestions by reviewers in the two rounds, I think that they improve your aticle.

Reviewer 1 ·

Basic reporting

I am satisfied that the authors have addressed my comments and the paper is suitable for publication.

Experimental design

n/a

Validity of the findings

n/a

Additional comments

n/a